# Institutional Quality, Trust in Institutions, and Waste Recycling Performance in the EU27

Andrea Pronti [1,2,*] and Roberto Zoboli [1,2]

1    DISEIS—Department of International Economics, Institutions and Development, Catholic University of Sacred Heart (Milan), Via Necchi 5, 20123 Milan, Italy; roberto.zoboli@unicatt.it
2    SEEDS—Sustainability Environmental Economics and Dynamic Studies, Via Voltapaletto 11, 44121 Ferrara, Italy
*    Correspondence: andrea.pronti@unicatt.it

**Abstract:** This paper addresses the role of institutional quality and trust in institutions for the performance in waste recycling of the EU27 countries. While survey-based works have highlighted the role of these factors for waste recycling attitudes and performance at the micro level, econometric analyses of recycling in Europe at country and regional levels have mostly looked into the role of waste policies, and not the role of institutional factors, in driving progresses in waste recycling. This paper tries to fill this gap through a panel econometrics analysis of recycling rates of municipal solid waste at the national level for the EU27 countries for the period 2005–2020. The proxies for institutional quality and trust in institutions, as the variables of interest, are introduced into a model that includes controls on a set of socio-economic variables, and on a set of EU waste policy variables, in particular the Waste Framework Directive (WFD) and its revision, and the first Circular Economy Action Plan of 2015 (CEAP 2015). Our results support the hypothesis that the quality of institutions can influence waste recycling performance. Moreover, our results provide evidence on the negative role of institutional trust on recycling rate at country level. Similarly, we find that the EU WFD and CEAP 2015 have been significant in driving recycling performances; the latter finding, however, is a necessary condition in appropriate institutional and socio-economic environments at the national level.

**Keywords:** waste recycling; quality of institutions; trust in institutions; circular economy action plan

## 1. Introduction

The EU is pursuing a large-scale sustainability transition through the European Green Deal (EGD), which is driven by the target of achieving the Net Zero of greenhouse gases emissions by 2050 [1]. This ambitious challenge has been confirmed by Next Generation EU (NGEU), the European post-pandemic recovery program, which includes specific constraints on allocation of funding to climate objectives [2]. Together with the focus on climate change, both the EGD and NGEU encompass the decoupling of economic growth from resource use, in which waste management and circularity have a key role, in particular through the new Circular Economy Action Plan 2020 embodied into the EGD strategy.

The circular economy paradigm is opposed to the predominant paradigm of the linear economy 'take, make, and dispose' and focuses on closing the cycle of materials and energy through the structural change of the whole production system [3,4]. The current linear economic model risks exceeding the environmental capacity to absorb the externalities and by-products of the economy and the regenerative capacity to produce enough materials and ecosystem services to sustain further development. The alternative is a shift to a more circular system, in which the decoupling of the economic activities is also achieved by reducing waste generation and increasing the reuse and recycling of materials, thus also benefitting ecosystems through reduced anthropogenic pressures from the extraction of virgin materials (e.g., loss of biodiversity, loss of ecosystem services).

Circularity can be achieved through increased material efficiency by using fewer resources for the same output, or reducing waste per unit of output, or reusing and recycling secondary materials avoiding the extraction of virgin materials and pressures on the ecosystem [5,6]. Key aspects of circularity are technological enhancement, environmental policies, and industrial changes, together with social participation in sustainable consumption behavior and minimized waste disposal.

Waste management is of primary importance within the circularity paradigm: it increases the amount of secondary material reintroduced into the economy, thus closing the material loop and avoiding the extraction of primary material and its impacts. Therefore, increasing recycling rates is at the very core of circularity progresses in accordance with the EU's main objectives of sustainability transition.

Municipal solid waste (MSW) amounts to around 10% of total waste generation in European countries [7–9]. Within the waste management systems of EU27 countries, recycling has gained a prominent role during the past few decades. On average, a European citizen produces 534 kg of waste of which 220 kg are recycled, 141 kg are treated in incinerators, and the remaining 173 kg are landfilled. However, even though all EU countries share a common legal EU framework that is pushing them to converge to the highest level of circularity, they show heterogeneous recycling performances within their own national and regional waste management models [10,11].

Many authors have investigated the role of EU and national policies, demographic factors, and socio-economic aspects as main drivers of waste recycling performances at national or regional levels (see [12–14]). While these analyses have often explored the role of specific waste policies for the observed changes in waste management in the EU countries, the factor which often has been overlooked is how the quality of institutions and the trust in institutions can influence waste management trends, and in particular the pattern of recycling activities. While it is expected that specific waste policies, for example landfill bans, can have a specific direct role in triggering positive changes in waste management, these policies are in any case implemented within national and local economic and social frameworks, in which the specific institutional environment and its capacities can have a critical role in the effectiveness of waste and recycling policies.

Institutions are the 'rules of the game' of a society and are important because they can push individual behavior towards collective actions that would not take place without them. Institutions enhance the cooperation of citizens through monitoring, coercion, and sanction systems that can prevent market failures or policy failures. The quality of institutions is linked to good governance with important impacts on socio-economic interactions within the society [15]. Moreover, trust in institutions, considered as the citizen's perception that these institutions can be trusted, can increase cooperation and participation in collective actions because citizens are confident that free-riding behaviors will not occur due to the presence of trustable institutions [16]. Institutions and institutional trust can affect how citizens participate in and cooperate within society, also determining the socio-economic performance of a country. Therefore, the impact of institutions on waste management and recycling is worthy of interest [17].

Some authors have analyzed the role of institutions in waste management with micro analyses using survey data [16,18] or sub-regional analysis in a single country [19]. All those studies have provided evidence on the effect of institutions on the recycling attitudes of citizens, but cross-country studies at the European level analyzing the effect of institutions on recycling are still lacking.

The aim of this paper is to investigate the role of the quality of institutions and institutional trust in the recycling performance of MSW in the EU27 by using an econometric approach to a cross-country panel dataset from Eurostat from 2005 to 2020. We aim to fill the gap in the literature by providing an econometric exercise on the main socio-economic determinants and the role of institutions in the MSW recycling rate of the EU27 countries along a time frame of 15 years.

The paper is structured as follows. Section 2 provides a background on recycling in the EU27 and a literature review on the quality of institutions and recycling. Section 3 provides the description of the data and method used in this study. In Section 4, the main results are presented and discussed in Section 5. The concluding remarks are presented in Section 6.

## 2. Background

### 2.1. Waste Policies and Recycling Performance in the EU27

Waste policies are among the oldest European policies for the environment. The first Waste Framework Directive of 1975 introduced the general approach to waste management based on the Waste Hierarchy, which gives the highest level priority to prevention and reuse (or preparation for reuse), followed by material recycling and then energy recovery, with landfill classed as the least preferred option (see Zoboli et al. [20] for a discussion). Important steps of the EU waste policies took place in the 1990s and early 2000s with the directive on packaging waste (1994) and the landfill and incineration directives (1999), together with the other directives on specific flows of waste (ELV, WEEE, and batteries) [21,22]. In the past two decades, the EU waste policies have been largely aimed at reinforcing the existing legal framework and at introducing more ambitious targets for the whole waste management system. Two major policy steps in the past fifteen years have been the new Waste Framework Directive of 2008, which was amended in 2018, and the First Circular Economy Action Plan (CEAP) of 2015, updated in 2020 within the European Green Deal package.

The new EU Waste Framework Directive (WFD) of 2008 updates the basic concepts and criteria of waste management policies and regulations, such as definitions of waste, recycling, and recovery to reflect decades of waste policy implementation and outcomes. The WFD also confirms the principles of Waste Hierarchy [23,24]. The aim of the WFD is to promote strategies for waste prevention prioritizing the reduction, reuse, and recycling of waste over disposal in landfill [25]. This policy vision and strategy was reinforced in 2015 with the First Circular Economy Action Plan (CEAP) [26]. While framed in the new dominating paradigm of the circular economy, the CEAP 2015 confirms the strategy of reducing the production of waste while closing the loops of material use in the EU economy [26]. Recycling is then a key strategy to move from a linear to a circular economy, and the recycling rate can be considered as a good proxy of measurement to assess the closure of the material loop within the economy [27,28]. In 2018, the WFD was amended by introducing new definitions and new targets for the recycling and reduction of municipal solid waste, together with other new provisions, for example, on the end-of-waste and on Extended Producer Responsibility (EPR). In particular, the preparation for reuse and the recycling of municipal waste should be increased to a minimum of 55% by weight by 2025, to a minimum of 60% by 2030, and to a minimum of 65% by 2035 [29]. The new CEAP (2020) focuses on product design for waste prevention, the extension of EPR to new sectors, e.g., single-use plastics, reuse, and recycling in the textile sector, and the improvement in the performance of secondary materials markets [20,30].

In general, the average recycling rate of MSW as a share of the total managed MSW in the EU27 countries has grown steadily over the last 25 years, from 12.3% in 1995 to 39.7% in 2020. Figure 1 illustrates the main waste management patterns in the EU27 countries. The average landfill rate of MSW experienced an important reduction in the past 25 years, from 70.2% of the total MSW in 1995 to 32.9% in 2020, while the average incineration rate has increased, but at a relatively slower rate compared to the recycling rate, reaching 32.9% of the total MSW in 2020.

Figure 2 shows the changes in the recycling rate for all EU27 countries from 2000 to 2020 and highlights important differences among countries in terms of recycling performances: only eight countries in 2020 recycled more than 50% of their total MSW production (Germany, Austria, Slovenia, Netherlands, Belgium, Denmark, Luxembourg, and Italy), thirteen countries recycled between 45% and 30% of their MSW (Lithuania, France, Slovakia, Finland, Ireland, Latvia, Poland, Sweden, Spain, Bulgaria, Croatia, Czechia, and

Hungary), and six countries recycled less than 20% of their MSW (Estonia, Portugal, Greece, Cyprus, Romania, and Malta).

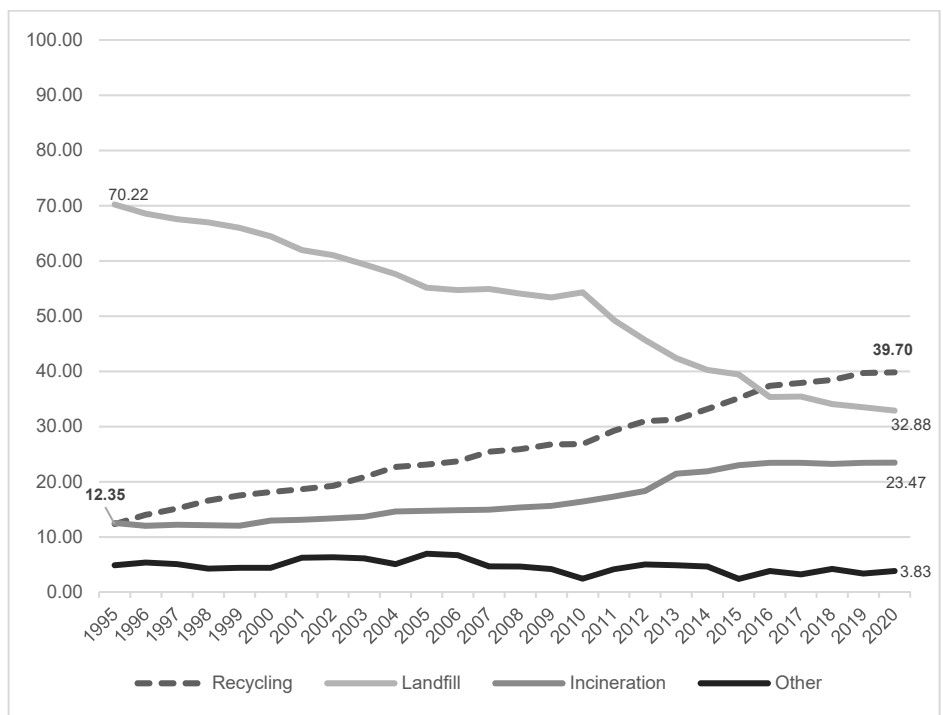

**Figure 1.** MSW treatment rate by category of treatment. Source: Authors' elaboration from Eurostat data.

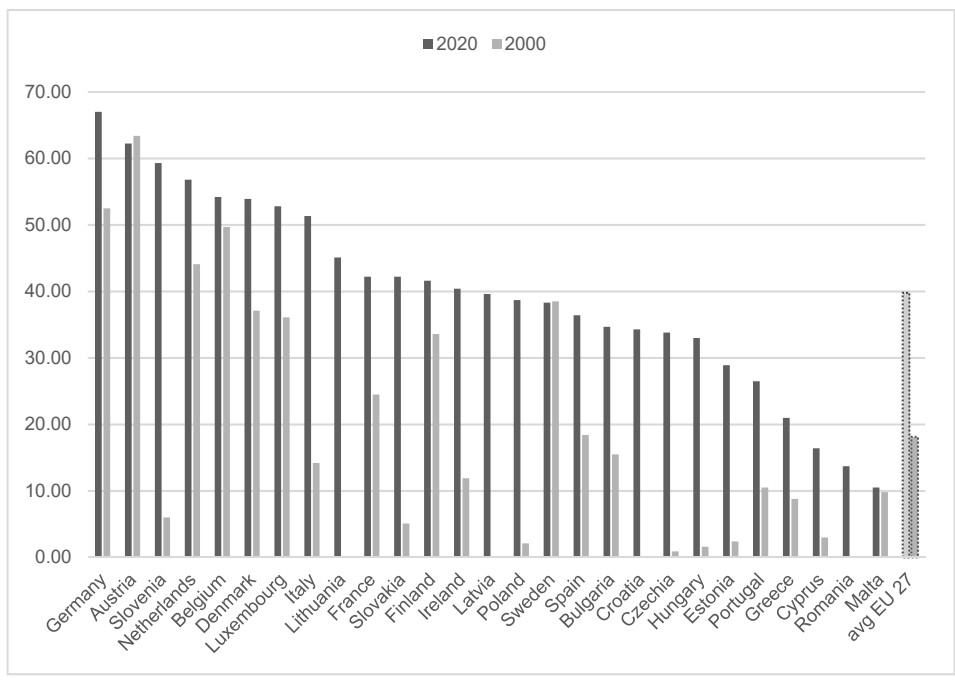

**Figure 2.** Recycling rate of MSW in the EU27 countries in 2020 and 2000 (in order of recycling rate in 2020). Source: Authors' elaboration from Eurostat data.

European countries, despite the common legal framework that includes binding targets of waste management performance to all EU members, have shown important heterogeneities in terms of recycling patterns. This has been due to the different strategies

for the diversion of MSW from landfills adopted by EU members during the past thirty years in which the landfilling, material recovery (mainly recycling and composting), and incineration of MSW were guided by different national and local structural socio-economic factors (i.e., the stage of economic growth, demographic and social aspects, investments in specific technologies creating lock-in effects, etc.) [10,13,14]. These differences in national, and local, patterns and performances are due not only to structural factors, but also to the specific institutional environment in which the policy impulses were received and processed to achieve the requested changes of the MSW management system. This role of the 'institutional environment' is the specific object of the rest of this paper.

### 2.2. Quality of Institutions, the Environment, and Waste Management

An important branch of economic research has studied institutional quality as a determinant of economic performance, highlighting the role of governance and institutions as important accelerators of economic development [31–38].

The quality of institutions is strictly related to good governance because it can reflect the ways in which authority is exercised in a country, including the political process by which governments are selected and monitored, their ability to implement appropriate policies, and the way they govern socio-economic interactions between the members of a country. Kauffman et al. [17] define 'governance' as "*(a) the process by which governments are selected, monitored and replaced; (b) the capacity of the government to effectively formulate and implement sound policies; and (c) the respect of citizens and the state for the institutions that govern economic and social interactions among them*".

Institutions are a fundamental factor of good governance and can be interpreted as the "rules of the game" of a society. They work as external systems of control that can drive the decision of individual members of the society toward the overall benefit for the society itself [15,39]. Formal institutions shape the relationships among individuals and reflect the government structure of a society [37,40], with important impacts on political, social, and economic relationships within the society [15].

Studying the effects of institutions on economic processes can be difficult because finding effective proxies for institutional quality is not an easy task. Governance is connected to various aspects which are not always observable, such as the rule of law (e.g., the enforcement of property rights), political stability (e.g., riots, violence), characteristics of political regimes (e.g., elections, constitutions, executive powers), social capital (e.g., civic participation), the control of crime and corruption, and other socio-cultural characteristics (e.g., income distribution, ethnicity, religion diversity, historical background) [41]. A robust approach is the one proposed by Kauffman et al. [17], who suggested a way to measure the quality of governance by articulating institutional quality around six dimensions that can have different impacts on growth and other socio-economic indicators: (1) voice and accountability; (2) political stability and absence of violence and terrorism; (3) government effectiveness; (4) regulatory quality; (5) rule of law; and (6) control of corruption [42].

'Trust in institutions' can be defined as the perception of and confidence of citizens in the credibility, fairness, competence, and transparency of institutions. Institutional trust is experiential in the sense that it depends on people's experiences with representatives of institutions (e.g., public employees such as bureaucrats and police officers) [43]. It is evident that general social trust, institutional trust, and institutional quality are correlated with important feedback effects on each other [39,43].

Differences among countries in the quality of institutions can lead to different outcomes in economic performance (i.e., a weak institutional framework can increase transaction costs, limiting the efficiency of markets, business, investments, and technological innovation) [41,44]. Therefore, the quality of institutions can also influence waste management at the country level.

'Institutional quality' and 'trust in institutions' can be even more important in waste management if recycling is considered as a special case of large-scale collective action dilemma or social dilemma [45]. A social dilemma can occur when in a collective action

situation (e.g., the provision of a public good, the internalization of externalities, or the management of the commons), the payoff of individuals for refusing action is higher than the cooperative actions, regardless of what other members do [45]. This leads to a final outcome of market failure depending on the type of problem (e.g., the public good is not provided, the externality is not internalized, or the commons collapse). Therefore, all individuals receive a lower payoff if they all refuse than if they all cooperate, but there is not a mechanism that allows the solution of the 'prisoner's dilemma' because the individual pay-off is larger than the social utility gained by the solution of the collective action problem.

In this framework, recycling takes the form of a public-good provision problem in which the successful provision of the public good itself (or of the shared commons) depends on the cooperation of a large number of individuals. In fact, individual participation in the schemes of advanced separate collection for recycling can influence the overall outcome of the recycling process: individual decisions, which depend mainly on individual incentives (i.e., economic, social, and other types of incentives), can affect the final level of total material recycled by the society as a whole [16,46]. People cooperate in collective actions if they have some confidence that others will also cooperate, and this may partly depend on general social trust, which can be considered as the expectations on others and on the reliability of their actions when there is no or little information about them [43]. In some cases, general social trust alone cannot guarantee the success of the social dilemma when the number of participant is large [47].

The higher the number of actors involved in a collective action problem, the higher the chances to end up in a situation far from the best social solution, as in the collective action problem explained by Hardin's '*Tragedy of the Commons*' [48]. The more actors are involved, the greater the demand for a third-party enforcer (e.g., the state or a lower level of government) to coordinate and facilitate collective actions, typically through the use of economic incentives, regulations, and control activities [18,39].

The introduction of third parties into collective action problems can prevent the failure of public good provision by increasing general trust in other members as well as trust in the institutional framework that enables the management of the collective action problem itself [49]. In the case of recycling, general trust in the institutions that manage the recycling process can increase the compliance and cooperative behavior of individuals by reducing incentives of free-riding through coercion, monitoring, and enforcement activities [16,18]. In other words, non-cooperation incentives which can increase defections from collective actions are lower with the participation of institutions in the process due to the general idea that institutions can manage and coordinate the contribution of all citizens to the public good.

Various studies highlighted the effect of institutional quality as a driver of good environmental performances of countries and regions, considering mainly air pollution and the management of natural resources [50–59], but few studies have focused on waste management.

Some authors specifically investigated the effect of the 'quality of institutions' and 'institutional trust' on the recycling behavior of citizens, finding that the trustworthiness of institutions increases cooperation in collective action problems, and in those cases, increasing participation in recycling activities. These studies have been mainly based on micro-level surveys.

Sønderskov [47] analyzed whether general social trust can determine recycling, finding evidence that people characterized by higher generalized social trust recycle more than those with low trust, and identifying how in large collective action problems, the idea of generalized social trust increases the level of cooperation when individuals believe that others will do the same.

Rompf et al. [16] tested the relationship between institutional and social trust and the recycling attitudes of citizens, also focusing on the interactions between these two components of trust, and identifying a different process of reciprocity in determining coop-

erative behavior. They found that an increase in institutional trust (which increases trust in the punishment of free-riders) prompts citizens toward an automatic norm-compliant behavior, affecting cooperation in recycling as a public good [16]. They also showed that when trust in institutions is high, the individual private costs and benefits do not affect the participation in recycling as a collective action.

Harring et al. [18] analyzed the link between institutional quality and trust and individual self-reported recycling behavior in different European countries, using cross-sectional survey data at the micro level. Moreover, the authors tested the hypothesis of a curvilinear relation between the self-reported recycling activities of citizens and their declared trust in institutions, which indicates a negative relationship between institutional trust and recycling. They argued that institutional trust at its highest levels, such as in developed countries, could affect cooperation negatively, because above certain high levels of trust, individuals no longer cooperate with the collective action problem. They do that by passivity or rational calculations, as their contribution appears to be less important since the state is assumed to take care of the public good regardless of individual actions [18]. In their empirical work, the researchers found that institutional quality, general social trust, and institutional trust all strongly increase recycling behavior, but failed to show an inverse relation between institutional trust and recycling activities.

Argentiero et al. [19] investigated the relationship between social trust and the quality of institutions on recycling behavior in Italy using data at NUTS 3 level to exploit the strong heterogeneity among Italian provinces in terms of institutions and social capital They consider social trust as a proxy of social cohesion and social capital. Moreover, they studied the interactions between social trust and the quality of institutions to consider potential mediating factors in influencing recycling attitudes. They found that both social trust and the quality of governance positively affect recycling, but their results also revealed a decreasing marginal effect of social trust when institutional quality is high, with a strong degree of substitution between trust and the quality of institutions [19]. This means that social trust for collective action is more important where institutional quality is low.

## 3. Methods and Data Description

### 3.1. Research Hypotheses

Our main interest is to understand and measure the effect of the 'quality of institutions' and 'institutional trust' on the recycling rate at country level for the EU27. Previous studies based on surveys, as reviewed above, confirmed a positive relation between institutional quality and recycling, which implies that the institutional environment positively affects the participation in and cooperation of households with recycling activities [16,19]. Our main hypothesis follows those findings and econometrically tests them for the EU27:

**H1.** *The stronger quality of institutions increases the level of recycling in the EU27 countries.*

This would imply a positive sign in the regression of the proxy for the quality of institution.

The study of Harring et al. [18] added institutional trust to the analysis of determinants of recycling behavior, assuming an unknown relation between the two variables and hypothesizing that a negative effect may also occur. This is explained in relation to a less civic participation when institutional quality is high because people believe that their contribution is useless since institutions 'take care of everything'. We partially follow their approach in defining our second hypothesis to be econometrically tested:

**H2**. *Institutional trust influences the recycling rate in the EU27 countries, but the direction of the effect is uncertain as it can be either an increase or a decrease in the recycling activities of citizens.*

This implies that a statistically significant coefficient of the proxy for institutional trust is required to test H2, but the sign of the coefficient cannot be defined a priori.

### 3.2. Data Description

All our data are from waste and socio-demographic datasets of Eurostat [60,61], and the time frame of analysis is 2005 to 2020 to circumvent missing data in many control variables in the years before 2005 in Eurostat data.

Our dependent variable *RR* is the recycling rate, measured as the total MSW recycled to the total MSW produced in each year considered in our analysis. Our key independent variables are *QI* and *IT*. The former (*QI*) is the quality of institutions, measured using the indicator on Government Effectiveness (GEE) from the World Governance Indicator (WGI) of the World Bank [42]. The WGI provides five other main composite indicators following the definition of the quality of institutions, as identified by Kaufmann et al. [17], namely, voice and accountability; political stability and absence of violence/terrorism; regulatory quality; rule of law; and control of corruption.

We focus only on the GEE indicator because all WGI indicators are strongly correlated to each other, thus preventing the possibility to use all of them together. The GEE indicator captures the citizens' perceived quality of public services, civil service, policy formulation, and implementation, the degree of independence from political pressures, and the credibility of the government's commitment to such policies [42]. Even if the WGI database provides a large selection of quality of institution indicators, we focused only on GEE because it is the closest indicator to be adapted to recycling and waste management. Furthermore, as the WGI indicators are set at world level, indicators other than the GEE (i.e., voice and accountability; political stability and absence of violence/terrorism; rule of law; and control of corruption) do not show great heterogeneity among European countries, since all EU27 countries have already converged towards high institutional quality standards for those aspects.

Our second key variable is institutional trust (*IT*) from Eurostat. Based on Eurobarometer surveys at a national level, it measures the confidence among citizens in a set of selected EU institutions (i.e., the European Parliament, the European Commission, and the European Central Bank). The variable represents the percentage of people positively declaring to trust in EU institutions on a three-grade answer ('tend to trust', 'tend not to trust', and 'don't know' or 'no answer') in each EU27 country. The institution we selected is the EU Parliament because it is the only elected institution among the three.

We identified the controls to be used in our model specification following previous analyses of recycling and waste management [7,12–14,24]. The data are from the Regional Eurostat database, and all the variables are registered annually considering the time frame 2005–2020. Our controls are as follows:

- Household size (*HH Size*). The dimension of the household can influence the level of waste generated and the amount recycled. Larger families can have more difficulty in recycling because of the higher amount of waste produced. The variable measures the average household size at country level.
- Low education (*Low Edu*). Recycling is expected to increase with higher levels of education which can influence more participation in large-scale collective actions because of more civic engagement or environmental concern. A higher number of citizens with low education can increase the level of non-compliance in recycling. This variable measures the percentage of 25–64 population with an education level lower than secondary (lower than primary, primary, or lower than secondary education) at country level.
- Immigration. A higher level of immigration is expected to reduce the level of recycling by several factors, for example, the unstable dwelling conditions of the immigrants; a low level of language comprehension; or the adherence to a traditional scheme of waste management, e.g., a culture of origin which does not consider recycling. The variable measures the total amount of immigrants resident in the country, using a natural logarithm to reduce the skewness of the distribution.
- Tourism. Recycling performance can be influenced by tourism flows in different ways, although the relationship between the two variables is not well defined. For example,

visitors may not be interested in participating in recycling activities because it is an action that requires effort in their free time, or they may not participate in recycling because they do not know how to comply with local recycling rules (e.g., the type of bins, or the type of selection of specific waste). This may reflect a negative sign of the tourism proxy in the regression. On the contrary, tourism flows may increase recycling activities due to the increased focus of local authorities on pro-environmental behavior, as tourists may have pro-environmental preferences, or they just prefer clean environments. Nevertheless, independently of the type or relation, in order to reduce distortions related to unobserved tourism activities, it is necessary to control for tourist activities. The variable measures the total number of tourist facilities (e.g., hotels, holiday and other short-stay accommodation, campsites, recreational vehicle parks, and caravan parks) per capita as a proxy for the total potential tourist accommodation, and it is calculated as the ratio between the total number of touristic establishments and the total population in a country.

- Population density (*Pop Density*). Several studies have already used this variable to control for economies of agglomeration and value of land that may substantially influence the cost of landfilling sites and therefore increase recycling activities because they reduce the overall cost of waste management [24]. Another aspect influenced by population density is the level of urbanization of a country which can directly affect the level of recycling through the economy of scale, integrated services, and the higher cost for other types of waste treatment. The variable measures the level of citizens per square kilometer living in a country, and we used the natural logarithm of population density to smooth the distribution.

- Age dependency ratio (*Age Dep*). The age structure of a country may influence the attitude towards recycling (e.g., younger citizens with greater environmental commitment may increase the overall recycling rate in a country). Although a clear relationship between the age of the population and recycling activities has not yet been established, it is necessary to control for this element as it could influence our estimation by biasing the results. The variable measures the age dependency ratio, as the percentage of the population in the non-working life stage divided by the population in the working life stage (i.e., the ratio of the population aged 0 to 19 and 65 or older to the population aged 20 to 64).

- Final consumption. One of the most important factors influencing waste generation is household consumption, which is also an important proxy for well-being and economic development, being strongly linked to gross domestic product per capita. Many other authors have used this variable in waste analysis, also considering its potential non-linearity in an environmental Kuznets curve hypothesis [12,13]. We follow this line of studies by adding the quadratic consumption term in our regression to consider non-linearity. The variable used is the household final consumption expenditure per capita at current prices at country level.

- Gini Index. Inequalities may affect recycling directly or indirectly. The first outcome can occur if different levels of recycling are due to inequalities within a country, which can result in differences in services provided (e.g., recycling services only in rich areas while poor areas are characterized by landfilling). The second outcome can depend on the overall institutional framework in poor areas, which can produce low recycling performances due to other institutional priorities (e.g., employment or welfare). The variable we employed is the Gini coefficient of equivalized disposable income before social transfers (pensions included in social transfers) expressed in a 0–100 range.

- High No-Waste Performances (*HNWP*). This variable can be interpreted as an indication of high performance in avoiding waste production, and it is used as a control for countries' profile and attitudes in limiting waste production. A country's recycling performance for MSW can be influenced by its idiosyncratic propensity to produce waste, which can be affected by various factors such as the consumption habits of the citizens, the overall circularity of the production system which reduces the parts of

goods becoming waste, or the pro-environmental behavior of the citizens. To consider these aspects, we used a dummy variable that takes the value 1 if waste production per capita is below the 10th percentile of the distribution of waste production per capita.

- Low No-Waste Performances (*LNWP*). This variable can be interpreted as an indication of a low performance in waste production per capita and it negatively mirrors the *HNWP* variable. We used a dummy variable taking the value 1 if the waste per capita is in the 90th percentile of the distribution. These last two variables (*HNWP* and *LNWP*) are used to control for lifestyle and efficient consumption management, and thus to consider the effect of the efficiency of consumption systems on recycling levels.

Table 1 provides a description of all the variables used in the econometric analysis.

**Table 1.** Descriptive statistics of the variables used in the econometric analysis.

| Variable | | Obs | Mean | Std. Dev. | Min | Max |
|---|---|---|---|---|---|---|
| Recycling rate | RR | 432 | 31.622 | 17.256 | 0 | 67.2 |
| Household size | HH Size | 432 | 2.447 | 0.268 | 2 | 3 |
| Low education | Low Edu | 432 | 23.355 | 14.574 | 4.6 | 74.8 |
| Log (Immigration) | Immigration | 432 | 10.819 | 1.384 | 7.27 | 14.267 |
| Tourism | Tourism | 432 | 16,920.605 | 34,806.283 | 157 | 226,855 |
| Log (Population density) | Pop Density | 432 | 4.647 | 0.901 | 2.793 | 7.395 |
| Age dependency | Age Dep | 432 | 63.9 | 5.239 | 52.3 | 80.2 |
| Final consumption | Consumption | 432 | 12,780.787 | 6339.909 | 2120 | 31,770 |
| Final consumption$^2$ | Consumption$^2$ | 432 | $2.034 \times 10^8$ | $1.841 \times 10^8$ | 4,494,400 | $1.009 \times 10^9$ |
| Gini Index | Gini Index | 432 | 48.633 | 4.519 | 37.2 | 61.6 |
| High No-Waste Performances | HNWP | 432 | 0.079 | 0.27 | 0 | 1 |
| Low No-Waste Performances | LNWP | 432 | 0.109 | 0.312 | 0 | 1 |
| Quality of institutions | QI | 432 | 1.098 | 0.583 | −0.36 | 2.354 |
| Trust in institutions (EU Parliament) | IT | 432 | 52.639 | 10.205 | 23 | 79 |
| Waste directive | WFD | 432 | 0.062 | 0.242 | 0 | 1 |
| Circular directive | CEAP | 432 | 0.062 | 0.242 | 0 | 1 |
| Revision targets | Revision | 432 | 0.062 | 0.242 | 0 | 1 |
| Trend 1 (2005–2007) | Trend 1 | 432 | 0.375 | 0.858 | 0 | 3 |
| Trend 2 (2009–2014) | Trend 2 | 432 | 1.312 | 1.994 | 0 | 6 |
| Trend 3 (2016–2017) | Trend 3 | 432 | 0.188 | 0.527 | 0 | 2 |
| Trend 4 (2019–2020) | Trend 4 | 432 | 0.188 | 0.527 | 0 | 2 |

*3.3. Econometric Strategy*

To test H1 and H2, we used a standard econometric panel data approach with a fixed effects model to consider unobserved heterogeneity and the potential endogeneity due to unobserved time-invariant variables which may bias our estimation [62]. To cope with potential autocorrelation and heteroskedasticity in our data, we used clustered standard error at country level [63,64]. Our main specification is outlined in Equation (1).

$$RR_{i,t} = \alpha + \beta_1 QI_{i,t} + \beta_2 IT_{i,t} + \sum_{m=1}^{k} \beta_m x_{m,\ i,t} + u_i + \varepsilon_{it} \qquad (1)$$

where *RR* is the recycling rate (MSW recycling over total MSW produced), *QI* is the quality of institutions, *IT* is the institutional trust, $x_m$ are other control variables, $u_i$ is the individual fixed effect, $\varepsilon_{it}$ is the idiosyncratic error, $\beta i$ are the parameters to be estimated, *i* is the identifier for the country, *t* is the identifier for the year, and $\alpha$ is the intercept. Our dependent variable is *RR*, while our key independent variables are the 'quality of institutions' *QI* and 'institutional trust' *IT*.

In a second specification, we added two dummy variables to control for the European policy framework which can have affected the recycling rate in our time frame of analysis. We add a dummy variable for 2008 when the Waste Framework Directive was implemented,

for 2015 when the first EU Circular Economy Action Plan (CEAP) was introduced, and for 2018 when the major EU targets for MSW were revised to become more ambitious. Moreover, to further control for EU policies and targets, which are expected to affect the recycling rate, and to consider the expected lagged effects of policies (i.e., the policy produces effects after its introduction and not in the year of implementation), we added a set of trends within our time frame:

- Trend 1 from 2005 to 2007, to control for the years before the introduction of the WFD;
- Trend 2 from 2009 to 2014, to check the effect of the WFD implementation before the introduction of the first CEAP;
- Trend 3 after 2015, to check the effect of the first CEAP before the revision of its targets which occurred in 2018;
- Trend 4 to check the effects of the target revision for the years after 2018.

The time frame of our analysis does not allow to consider a possible effect of the New Circular Economy Action Plan of 2020.

The second specification is described in Equation (2), in which the dummies for the EU policies and the trends ($\tau_1$, $\tau_2$, $\tau_3$) were added to the previous specification.

$$RR_{i,t} = \alpha + \beta_1 QI_{i,t} + \beta_2 IT_{i,t} + \sum_{m=1}^{k} \beta_m x_{m,\,i,t} + u_i + WFD_{i,t} + CEAP_{i,t} + Revision_{i,t} + \tau_{1(2005-2007)} + \tau_{2(2009-2014)}$$
$$+ \tau_{3(2016-2018)} + \tau_{4(2019-2020)} + \varepsilon_{it} \tag{2}$$

Finally, we used the specification shown in Equation (3) to test the hypothesis of Harring et al. [18], which states that trust in institutions can have a negative effect at high levels of waste management performance, since the individual contribution may be felt to be unnecessary as the institution itself is assumed to take care of the waste regardless of individual action.

$$RR_{i,t} = \alpha + \beta_1 QI_{i,t} + \beta_2 IT_{i,t} + \sum_{m=1}^{k} \beta_m x_{m,\,i,t} + u_i + WFD_{i,t} + CEAP_{i,t} + \tau_1 + \tau_2 + \tau_3 + \tau_4 + \beta_{k+1} HNWP_{i,t}$$
$$+ \beta_{k+2} HNWP_{i,t} * IT_{i,t} + \beta_{k+1} LNWP_{i,t} + \beta_{k+2} LNWP_{i,t} * IT_{i,t} + \varepsilon_{it} \tag{3}$$

The last specification shown in Equation (3) is defined as before, but in this specification, two interaction terms were added to better identify how institutional trust operates in affecting recycling attitudes. We used the interaction between institutional trust and the dummies of high and low waste production performances (*HNWP* and *LNWP*), indicating the mediating effect of institutional trust with countries that have both high and low levels of waste production performances. By doing this, we could identify the effect of institutional trust on countries in the 10th and the 90th percentile in terms of 'no-waste' performance: this can suggest whether institutional trust might negatively affect recycling at high levels of 'no-waste' performance. To confirm this, we expected that the interaction between *IT* and *HNWP* (i.e., countries that have a high level of 'no-waste' performance) would be statistically significant and negative. This effect should not be relevant for the interaction between *IT* and *LNWP* (i.e., the interaction between trust in institutions and the dummy of low 'no-waste' performance countries should not be statistically significant).

## 4. Results

The results of our analysis are shown in Table 2. The quality of institutions (*QI*) using WGI's government effectiveness indicator, is always statistically significant alone (90% level) and in combination with *IT* (95% level), with a positive coefficient indicating that a high level of institutional quality increases the recycling rate (Column 2 to 4). In terms of marginal impacts, the quality of institutions increases the recycling rate by 7.6% and 8.3% % for the specifications in columns 2 and 3 (respectively, in model 2 without *IT* and model 4 with *IT*).

**Table 2.** Results of the econometric analysis. Effects of the quality of institutions (QI) and institutional trust (IT) on the recycling rate (RR).

| | (1) | (2) | (3) | (4) | (5) |
|---|---|---|---|---|---|
| **Variables** | **Model 1** | **Model 2** | **Model 3** | **Model 4** | **Model 5** |
| HH Size | −11.75 | −9.988 | −9.632 | −7.572 | −8.012 |
| | (−1.519) | (−1.323) | (−1.282) | (−1.028) | (−1.090) |
| Low Edu | 0.125 | 0.115 | 0.121 | 0.109 | 0.0915 |
| | (0.612) | (0.575) | (0.616) | (0.575) | (0.484) |
| Immigration | 0.217 | 0.0732 | 0.528 | 0.396 | 0.552 |
| | (0.217) | (0.0731) | (0.546) | (0.413) | (0.578) |
| Tourism | 0.000184 *** | 0.000180 *** | 0.000187 *** | 0.000183 *** | 0.000183 *** |
| | (7.528) | (8.316) | (8.194) | (8.507) | (9.031) |
| Pop Density | −18.49 | −10.35 | −25.38 | −17.09 | −19.68 |
| | (−0.758) | (−0.428) | (−1.106) | (−0.761) | (−0.872) |
| Age Dep | 0.0571 | 0.124 | 0.0133 | 0.0819 | 0.113 |
| | (0.162) | (0.373) | (0.0397) | (0.263) | (0.359) |
| Consumption | 0.00510 *** | 0.00496 *** | 0.00499 *** | 0.00483 *** | 0.00460 *** |
| | (3.768) | (3.803) | (3.987) | (4.143) | (4.107) |
| Consumption^2 | $−1.18 \times 10^{-7}$ ** | $−1.15 \times 10^{-7}$ *** | $−1.02 \times 10^{-7}$ ** | $−9.80 \times 10^{-8}$ *** | $−9.22 \times 10^{-8}$ *** |
| | (−2.661) | (−2.784) | (−2.643) | (−2.874) | (−2.804) |
| Gini Index | −0.0893 | −0.0649 | −0.132 | −0.109 | −0.0747 |
| | (−0.692) | (−0.487) | (−0.991) | (−0.827) | (−0.566) |
| HNWP | −1.335 | −1.657 | −0.684 | −0.984 | 18.90 ** |
| | (−0.641) | (−0.848) | (−0.332) | (−0.509) | (2.659) |
| LNWP | −2.490 * | −3.170 ** | −2.792 ** | −3.549 *** | −3.874 |
| | (−1.952) | (−2.382) | (−2.387) | (−2.982) | (−0.531) |
| QI | | 7.653 ** | | 8.271 ** | 8.374 ** |
| | | (2.356) | | (2.422) | (2.462) |
| IT | | | −0.169 ** | −0.181 ** | −0.165 ** |
| | | | (−2.285) | (−2.623) | (−2.300) |
| HRP*IT | | | | | −0.353 ** |
| | | | | | (−2.739) |
| LRP*IT | | | | | 0.00264 |
| | | | | | (0.0196) |
| WFD | −2.253 *** | −1.970 ** | −2.380 *** | −2.084 ** | −1.962 ** |
| | (−3.002) | (−2.681) | (−3.053) | (−2.755) | (−2.724) |
| CEAP | 4.745 *** | 4.852 *** | 2.642 * | 2.604 ** | 2.286 * |
| | (3.696) | (3.821) | (1.985) | (2.080) | (1.870) |
| Revision | 4.688 *** | 5.030 *** | 3.589 *** | 3.878 *** | 3.630 *** |
| | (3.405) | (3.596) | (3.174) | (3.419) | (3.166) |
| Trend 1 | −0.600 ** | −0.502 ** | −0.361 | −0.238 | −0.229 |
| | (−2.738) | (−2.148) | (−1.650) | (−0.911) | (−0.840) |
| Trend 2 | 0.415 * | 0.430 * | 0.114 | 0.109 | 0.0424 |
| | (1.871) | (2.053) | (0.482) | (0.493) | (0.198) |
| Trend 3 | 3.292 *** | 3.450 *** | 2.421 *** | 2.528 *** | 2.357 *** |
| | (4.553) | (4.883) | (3.718) | (4.027) | (3.712) |
| Trend 4 | 3.388 *** | 3.803 *** | 3.028 *** | 3.449 *** | 3.280 *** |
| | (3.196) | (3.631) | (3.166) | (3.692) | (3.540) |
| Constant | 95.74 | 42.72 | 132.0 | 77.38 | 86.65 |
| | (0.720) | (0.320) | (1.074) | (0.631) | (0.703) |
| Observations | 432 | 432 | 432 | 432 | 432 |
| R-squared | 0.647 | 0.660 | 0.661 | 0.675 | 0.682 |
| Number of Id | 27 | 27 | 27 | 27 | 27 |

Robust t-statistics in parentheses. *** $p < 0.01$, ** $p < 0.05$, * $p < 0$.

Institutional trust (*IT*) is also significant in all specifications with a statistical significance level of 95% (model 3 and 4), but the effect of this variable is negative in all specifications, with a magnitude of −0.17 in the specification without *QI* (model 3) and −0.18 with QI (model 4). This result indicates that institutional trust reduces the recycling

rate at the national level, with a reduction between $-0.17$ and $-0.18$ for each additional percentage of the population declaring to trust in the European Parliament, depending on the model specification. This suggests that increasing levels of institutional trust can have a negative effect at its highest level, thus reducing the recycling activities of citizens, as hypothesized by Harring et al. [18].

To confirm this, we ran an additional regression in which we inserted in the specification two additional elements which can be useful to identify how the effect of institutional trust operates in affecting recycling attitudes. The results of this econometric exercise are shown in model 5 where we added the interaction between institutional trust and the dummies of high and low waste production performances (*HNWP* and *LNWP*).

In this last specification, the signs and magnitude of *QI* and *IT* remain stable and statistically significant (both at the 95% level with a magnitude, respectively, of 8.37 and $-0.16$), confirming their main effects on the recycling rate. Moreover, we found a statistically significant negative effect of the interaction between *IT* and *HNWP* with a magnitude of $-0.353$. This suggest that for a high level of 'no-waste' performance at country level, the effect of institutional trust may reduce the overall recycling performance. This result further confirms the hypothesis of Harring et al. [18] that, at a high level of institutional quality, an individual can turn away from cooperative behavior, producing a sub-optimal outcome in a large-scale collective action problem, because an excessive trust in the institutional system can boost non-cooperative attitudes. This is confirmed by the positive sign, the low magnitude (0.00264), and the statistical insignificance of the coefficient of the interaction term between institutional trust and *LNWP*.

Considering the control variables, *Tourism* is always statistically significant at 99% in all specifications, which means that the tourism economy can push towards higher rates of recycling, but the magnitude of the coefficient prevents us from giving this variable a relevant economic role for the recycling rate (the coefficient is a five-digit number). The effect of the household size (*HH size*) is negative, as expected, but the estimated coefficient is never significant.

Household final consumption is positive and strongly significant in all models (99%). It is interesting that the non-linear effect of household final consumption is confirmed since the quadratic value of the variable (*Consumption^2*) is negative, which indicates a decreasing marginal effect of household final consumption on recycling activities along an inverse U-shaped curve. Although the magnitudes of the two coefficients are low, the effect of final consumption on the recycling rate has to be considered in terms of the marginal increase in the recycling rate due to each additional euro spent on average on consumption by households in the EU27: each additional euro spent on consumption by an average EU27 household increases the recycling rate by 0.005%; when considering the non-linearity, the negative effect of the quadratic term on the marginal increase in the recycling rate due to consumption activities is negligible (8-digit coefficient).

All other controls showed no statistical significance. They contributed to the correct specification of the econometric models, but we do not comment on them.

Among the waste policy variables, all yearly dummies used for waste policies are statistically significant in almost all specifications. The coefficient of the WFD is negative, indicating that in 2008, a slight reduction in the recycling rate occurred, whereas in 2015, with the introduction of the CEAP, and in 2018, with the revision of the waste targets, the recycling rate increased since the coefficients are all positive. It should be noted that the variables we used are basically dummy variables for the years 2008, 2015, and 2018, in which the three policies were introduced; therefore, those results should be taken cautiously since there may have also been other aspects affecting the recycling rate occurring in that specific year. Clearer evidence on the effects of waste policies should be further studied using specific econometric analysis (e.g., difference in differences, matching estimations).

Nevertheless, additional evidence of the effects of waste policy implementation can be derived from the analysis of the coefficients of the time trends we used for the years between the introduction of each policy: they show stable signs in all specifications and

are statistically significant in many of them, thus being consistent with the results of the policy-introduction dummies.

These trends can be meaningful in interpreting the results for policies, because the actual effects of the introduction of waste policies are subject to lags and their possibly successful implementation in each country occurs with a delay, even just technical in nature, usually displaying their effects only in later years. These time lags can also reflect the quality of national and local institutions, as well as the possible financial constraints to investments in industrial capacity for waste management and recycling.

The first trend (2005–2007) is negative in model 1 and 2, confirming that in the years before 2008, the trend of *RR* was in slight decline. After the introduction of the WFD in 2008, the *RR* started to increase as it is shown by the positive coefficient of trend 2 (in model 1 and 2). This highlights that the WFD may have positively influenced the recycling activities in the EU27 countries. The most interesting effects are shown by trends 3 and 5, which are always statistically significant at 99% with a positive effect on *RR*, suggesting that both the introduction of the first CEAP and the 2018 revision of waste targets produced positive effects on recycling activities in the years following their introduction. Considering the marginal contribution of the first CEAP in the year following its first introduction, the *RR* increased by 2.53% in 2016 and 2017, respectively, while the revision of the waste targets further increased the *RR* by 3.5% in 2019 and 2020, respectively (see Model 4).

## 5. Discussion

Our findings confirm, with a different approach, the results of Rompf et al. [16], Harring et al. [18], and Argentiero et al. [19], who highlighted that institutional quality (i.e., good government) can increase waste recycling performance. This suggests that good and solid institutions can improve performance in solving large-scale collective action problems in public goods management such as the recycling of waste and other environmental issues. From our findings, it is clear that a higher level of institutional quality can increase the recycling performance in the EU27 countries. This might depend on a better organization of waste management, like collection, which is within the competences of the public sector in many countries, a higher level of control, punishment, coercion, and other incentives which can guide individual decision making toward cooperative actions, thus helping in solving collective action problems.

While the positive influence of institutional quality on recycling activities is fairly straightforward to understand, our findings on the effect of institutional trust are less clear. Our results indicate that institutional trust has a negative impact on the rate of recycling, suggesting that for a high level of institutional quality, institutional trust can reduce citizen participation in collective action problems, potentially limiting the social outcome when social dilemmas for the provision of public goods are at play.

Other authors have analyzed the effects of institutional trust on recycling activities, finding results that are different from those of our analysis. Sønderskov [47] found a positive effect of institutional trust as an increasing factor of social trust on recycling behavior. The focus of the author was more related to social trust, while the role of institutional trust was deepened in his further studies but not in relation with recycling attitudes. Rompf et al. [16] found a positive relationship between institutional trust and recycling behavior (i.e., attitude toward recycling); they also found a mediating negative effect with private benefits in recycling (i.e., a higher level of institutional trust reduces the personal benefit to compensate personal recycling costs) and a positive mediating effect with the cost of recycling (i.e., the effect of recycling costs on recycling attitudes decreases with an increasing level of institutional trust). The authors clearly argue that institutional trust, considered as trust in the reliability, effectiveness, and legitimacy of public institutions, has an overall positive effect on recycling activities, and conclude saying that improving the quality of institutions and the citizens' perception of them as trustworthy can increase individual incentives to solve the collective action dilemma applied to waste recycling activities [16].

Instead, our findings are in line with the hypothesis of Harring et al. [18], who argue that institutional trust is not just positive in influencing the cooperative actions of individuals, but conversely, in high-trusting societies, above certain levels of trust, a continuous positive relationship between institutional trust and recycling appears to be far from obvious.

The authors suggest that a strong faith in the state's ability to solve complex issues may make personal contribution to be perceived as less important, which may lead to uncooperative behavior at a certain level, the latter depending either on a rational decision not to cooperate or just on individual passivity.

The overall outcome is that citizens in high-trusting countries, where the quality of institutions is high, can reduce their personal contribution to collective actions in the presence of large-scale social dilemmas. The authors clearly stated a very interesting and plausible hypothesis, but in their empirical analysis, they did not find any evidence of that [18]. The results of our analysis confirm the argument of Harring et al. [18]: our results indicate that for a high-quality institutional environment, as in most EU27 countries, the effect of institutional trust on recycling participation can be negative. This is also confirmed when we combined the *IT* variable with the dummies of the highest and lowest deciles of the distribution of waste generation per capita, with, respectively, a positive and non-significant effect for a high level of waste generation per capita and a statistically significant negative effect for a low level of waste generation per capita.

This combination of results can suggest that, in achieving higher rates of recycling, administrative capacity, as reflected in the variable *QI*, is important, but public trust in institutions, as reflected in *IT*, due to the high level of quality achieved by the institutional system, may reduce the individual contribution to large-scale collective actions. In other words, the quality of institutions, and thus good administration, may be a sufficient condition for good recycling performance, but as discussed above, trust in institutions may reduce the overall effect of good administration on waste recycling.

Furthermore, our results highlight the effectiveness of waste policies introduced by EU institutions. In fact, our trend analysis shows that after the introduction of the WFD and CEAP, the average rate of recycling increased in the countries under study in the years after the implementation of these new waste policies.

Therefore, our analysis confirms that increasing the quality of institutions can improve environmental sustainability in a problem area, like waste management, in which the active contribution of citizens is fundamental for achieving the policy objectives. EU policies can be important in driving national policies in terms of sustainability and circularity, but strong and efficient national institutions can positively affect citizen participation to achieve high recycling performance of the EU27 countries. Then, improving good governance may increase the perceived quality of citizens helping to overcome a social dilemma which prevents circularity. This may apply to other environmental sectors in which citizen participation is important to achieve sustainability (e.g., pollution, climate change, and adaptation strategies), and further studies may also investigate in that direction. Considering potential individual defections deriving from a curvilinear institutional trust, when institutions are good, in order to improve citizen participation to collective actions, EU member states should work together to increase the level of individual cooperative action by increasing the pro-environmental behavior of citizens using both social stimuli (e.g., social involvement in the ecological transition process) and economic incentives (e.g., by using financial stimuli, such as pay-as-you-throw tariff schemes). We analyzed recycling activities, but as pointed out by Fellner et al. [65], recycling does not consider the effective reuse of secondary materials within a country, and therefore further studies should consider more comprehensive measures of circularity in which the reuse of goods and materials is also considered.

## 6. Conclusions

Looking at recycling as a large-scale collective action problem, in this paper, we analyzed the effect of the quality of institutions and the trust in institutions on recycling rate dynamics in EU countries. Our study uses cross-country data from Eurostat and WGI for a fifteen-year timeframe (2005–2020), with a panel data econometric approach.

Our findings support previous survey-based evidence developed at the micro and local level, in which it was highlighted that the quality of institutions can increase the social participation of citizens in recycling activities [18,19]. In our framework, the significance of institutional quality also suggests the role of good administration in providing sufficient waste collection and management facilities and infrastructure to increase recycling performance. On the other hand, our findings do provide evidence of the possible negative role of institutional trust on the recycling rate at country level, resulting from non-linear dynamic interactions between the quality of institutions and the trust in institutions. The quality of institutions and institutional trust are dynamically linked (i.e., the quality of institutions increases institutional trust) [43], but they can have two different and opposite effects on recycling. Further studies should consider measures to better disentangle the link between the two processes, as well as designing policies able to compensate for the decreasing individual participation in large-scale collective actions due to the high level of institutional trust with a high quality of institutions, both in general and for the environment.

Similarly, when considering the impact of specific EU policies within the timeframe of our analysis (WFD and CEAP), we see that they have been significant for increasing the recycling rate at the EU27 level. The stimulus from specific policies can thus be seen as a driver of recycling, as emerging from other analyses, but it is mainly because policies trigger processes that call for the work of appropriate institutional and socio-economic environments to achieve the policy-desired results. In a way, strong waste and recycling policies, while activating significant changes in waste management and circularity, cannot be effective if they do not find appropriate institutional and administrative systems in the implementation phase. Given that the latter is mainly a national or local matter, and the processes leading to good institutions are slow and systemic in nature, the design of EU waste and recycling policies should pay more attention to a range of enabling factors beyond ambitious targets and detailed regulations, like measures aimed at creating better markets for recycling and reuse, and a better waste management infrastructure [30].

**Author Contributions:** Conceptualization, A.P. and R.Z.; Methodology, A.P. and R.Z.; Software, A.P.; Validation, R.Z.; Formal analysis, A.P.; Investigation, A.P. and R.Z.; Data curation, A.P.; Writing—original draft, A.P.; Writing—review & editing, A.P. and R.Z.; Supervision, R.Z. All authors have read and agreed to the published version of the manuscript.

**Funding:** This research received no external funding.

**Informed Consent Statement:** Not applicable.

**Data Availability Statement:** Data are fully available from the online free sources cited in the references (Eurostat and WGI).

**Conflicts of Interest:** The authors declare no conflict of interest.

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
