# Peer review of "Institutional Quality, Trust in Institutions, and Waste Recycling Performance in the EU27"

_sustainability, doi:10.3390/su16020892_

Round 1

Reviewer 1 Report

Comments and Suggestions for Authors

Authors should have used the standard template and included  line numbers for an easier examination of the MS. Type of paper should be indicated (I assumed it to be article). Being an expert in waste management and not in economic and social policies, I had some difficulty to read the MS as its premises are not explained very clearly and used data have not been made explicit (in supplementary materials).

Comments on the Quality of English Language

English is fine. Minor corrections suggested.

Author Response

Dear reviewer 1,

thanks for your valuable contribution to improve our paper. We deeply modified the manuscript following all your suggestions. Specifically, here below we answer to what you pointed out in your revision:

Authors should have used the standard template and included line numbers for an easier examination of the MS. Type of paper should be indicated (I assumed it to be article). Being an expert in waste management and not in economic and social policies, I had some difficulty to read the MS as its premises are not explained very clearly and used data have not been made explicit (in supplementary materials).”

We modified the data section introducing two new variables used to improve the analysis. Moreover, we added a table in which are shown the descriptive statistics of the variables used in our econometric analysis. All our data are from Eurostat and freely accessible, in the references we cited the source with a URL to access easily to the databases. We added line numbers to improve the readability of the referees.

The template of the journal is not compulsory anymore, but we changed the reference style following the guidelines of the journal.

Finally, we explicit (as we did during the submission) that this paper is a research article.

We hope that the changes we made can respond to your requests.

Thanks again and kind regards.

The authors

Reviewer 2 Report

Comments and Suggestions for Authors

Thank you very much for your manuscript, I read it with interest.

Please check your language a bit. There are few issues with language clarity and readability throughout the main text and also in the abstract. For example, H1 is not stated correctly – but this is not only an issue of language but also of how hypotheses work. Please reformulate in a way that H1 would be clearer – it is “improved” quality of democracy? Or “better”, “stronger”, etc.? Also H2 is not clear, moreover, its second part is rather confusing – isn’t rejection of a hypothesis doing what the second part of your hypothesis claim? (That we do not know the direction?)

I am not sure about the variable recycling intensity and its relation to recycling rate. Please clarify this because from your analysis (p.12, third paragraph) it seems that these two are closely connected: “This indicates that the higher the size of the recycling activity of a country the higher the recycling rate”. This does not seems to be surprising (or interesting; contrary to what you claim in this paragraph) as the two seems to be interlinked – you need bigger industry to deal with higher shares or recycled material. So maybe there is an opposite direction of influence – first, to support more recycling, big industry is being developed to deal with expected amount of stuff to recycle (that comes later thanks to the existing capacity).

Your results claim that “On the other hand, the results of our analysis have not evidenced any statistical effects of the implementation of the WFD on RR neither in the year of implementation nor in the timeframe between the introduction of the CEAP”. Honestly, this is difficult to believe as these EU-wide rules brought stricter recycling rules to countries that had previously a bit different approach to recycling. Take Poland, for example – of many countries of Central and Eastern Europe. Their great progress in recycling (Figure 2) can be explained similarly as in progress in many other policies – pressure from EU and its rules. I am not saying that your analysis is incorrect, but please explain your results better. Because – to use Polish example again – their institutions could be only difficult to consider of the highest “quality” within the EU.

I would recommend dividing your “Results and Discussion” section into two as now you are mixing these two things and I believe it would be better to present your results first and then discuss them thoroughly. As of now, I think your discussion section need to be revised and upgraded a bit to include more discussion with existing literature (there is some, but I think you need more)

Comments on the Quality of English Language

There are only very few minor issues that can be easily fixed by the authors when they go through the text again. 

Author Response

Dear reviewer 2,

thanks for your valuable contribution to improve our paper. We deeply modified the manuscript following all your suggestions. Specifically, here below we answer to what you pointed out in your revision:

Please check your language a bit. There are few issues with language clarity and readability throughout the main text and also in the abstract. For example, H1 is not stated correctly – but this is not only an issue of language but also of how hypotheses work. Please reformulate in a way that H1 would be clearer – it is “improved” quality of democracy? Or “better”, “stronger”, etc.? Also H2 is not clear, moreover, its second part is rather confusing– isn’t rejection of a hypothesis doing what the second part of your hypothesis claim? (That we do not know the direction?)

Thanks for your suggestions, we have checked all the English and now the clarity and readability of the paper improved a lot more. Moreover, we modified H1 and H2 to improve their interpretations. We invite you to check if those are now in line with your expectations.

I am not sure about the variable recycling intensity and its relation to recycling rate. Please clarify this because from your analysis (p.12, third paragraph) it seems that these two are closely connected: “This indicates that the higher the size of there cycling activity of a country the higher the recycling rate”. This does not seems to be surprising (or interesting; contrary to what you claim in this paragraph) as the two seems to be interlinked –you need bigger industry to deal with higher shares or recycled material. So maybe there is an opposite direction of influence –first, to support more recycling, big industry is being developed to deal with expected amount of stuff to recycle (that comes later thanks to the existing capacity).

You are definitively right: recycling intensity and recycling rate are strongly correlated and therefore the previous results on the RI coefficient risked to tautological as you pointed out. In our previous analysis, we were focusing on recycling intensity to find a proxy describing the quality of the waste management systems across countries since this might influence the level of MSW recycling, but as you commented that was not correct way. Therefore, to consider this aspect we introduced two new dummy variables High No-Waste Performance (HNWP) and Low No Waste Performances (LNWP). HNWP takes the value 1 if the country is in the lowest decile of the waste production per capita distribution (i.e. it indicates that the country has a good performance in avoiding waste production), while LNWP takes the value 1 if the country is in the highest decile of the waste production per capita distribution (i.e. it indicates that the country has a bad performances in avoiding waste production). By using those variables we can control for structural or idiosyncratic factors which each country has in waste management. Without controlling for that, our results can be biased by this heterogenous unobserved effect. Doing this has improved drastically our results and we commented them in details in the results, discussion and conclusion sections. Now the level of the manuscript has greatly improved in our opinion.

Your results claim that “On the other hand, the results of our analysis have not evidenced any statistical effects of the implementation of the WFD on RR neither in the year of implementation nor in the timeframe between the introduction of the CEAP”. Honestly, this is difficult to believe as these EU-wide rules brought stricter recycling rules to countries that had previously a bit different approach to recycling. Take Poland, for example – of many countries of Central and Eastern Europe. Their great progress in recycling (Figure 2) can be explained similarly as in progress in many other policies – pressure from EU and its rules. I am not saying that your analysis is incorrect, but please explain your results better. Because – to use Polish example again – their institutions could be only difficult to consider of the highest “quality” within the EU.

Thanks for your comment. After changing our model specification with the introduction of HNWP and LNWP, we found an improvement in the recycling rate after the introduction of WFD and the CEAP. This is revealed by the positive effect in the trends after the introduction of the policies. The one-year dummy fro the introduction of WFD is still negative, but it can be understood considering that policy effects can be lagged and then become effective in the following years. This lagged effect can be confirmed by the coefficient of trend 2 (the years after the WFD introduction) which has a positive coefficient. Furthermore, trend 1 (indicating the years before the introduction of the WFD) is negative, indicating that a decrease in the recycling rate was taking place before the introduction of the WFD. Also trend 3 and trend 4 identified a positive influence of the introduction of CEAP on the recycling rate in EU27 countries. We commented our findings on the effectiveness of waste policies in the new discussion section and in the conclusions. We hope that now our results are in line with our expectations.

I would recommend dividing your “Results and Discussion” section into two as now you are mixing these two things and I believe it would be better to present your results first and then discuss them thoroughly. As of now, I think your discussion section need to be revised and upgraded a bit to include more discussion with existing literature (there is some, but I think you need more)

Thanks for your suggestion again. We followed your indication splitting the two sections up. In the discussion section we confronted our results with the existent literature in details.

We hope that the changes we made are in line with your requests.

Thanks again and kind regards.

The authors

Reviewer 3 Report

Comments and Suggestions for Authors

Dear Authors,

The manuscript titled “Institutional quality and waste recycling performance in the EU27” makes a valuable contribution to understanding how recycling works in the European Union in the last 15 years. This overview article is well-structured; moreover, empirical studies give valuable information about current problems on how to successfully guide waste policies inside environmental institutions. Introduction gives relevant observation about European ecosystem services efficiency. The literature review contributes to adapting novel approaches to educate and encourage society to change environmental governance 'closing the loop on waste materials' according to the CE Action Plan 2020. Data and researching methodology give valuable information to understanding environmental governance, which relate to the institution efficiency, how citizens respect its “society game roles”. Authors are selected measured variables (HH size, Low Edu, Immigration, Tourism, Pop Density, Age Dep., Final consumption, Gini Index, RI) and give comprehensive and relevant analysis about the current situation in the EU27. Before accepting the article, only two comments should be addressed:

1. The conclusion should be boosted, explain socio-economic performance, inside selected variables which are going to directly contribute recycling agendas.

2. Editing of the layout must be prepared in accordance with the rules of the Journal (the serial number of the reference is missing)

Best Regards,

Author Response

Dear reviewer 3,

thanks for your valuable contribution to improve our paper. We deeply modified the manuscript following all your suggestions. Specifically, here below we answer to what you pointed out in your revision:

1. The conclusion should be boosted, explain socio-economic performance, inside selected variables which are going to directly contribute recycling agendas.

  1. Editing of the layout must be prepared in accordance with the rules of the Journal (the serial number of the reference is missing)

We reviewed our analysis inserting two new variables. This improved drastically our results and we commented them in details in the results, discussion and conclusion section. Now the quality of the manuscript has greatly improved in our opinion.

Moreover, as you suggested, we changed the reference style following the guidelines of the journal.

We hope that the changes we made are in line with your requests.

Thanks again and kind regards.

The authors